# Down to the Bone: A Novel Bio-Inspired Design Concept

**DOI:** 10.3390/ma14154226

**Published:** 2021-07-28

**Authors:** Federica Buccino, Irene Aiazzi, Alessandro Casto, Bingqi Liu, Maria Chiara Sbarra, Giovanni Ziarelli, Laura Maria Vergani, Sara Bagherifard

**Affiliations:** Department of Mechanical Engineering (DMEC), Politecnico di Milano, Via La Masa 1, 20156 Milano, Italy; federica.buccino@polimi.it (F.B.); irene.aiazzi@asp-poli.it (I.A.); alessandro.casto@asp-poli.it (A.C.); bingqi.liu@asp-poli.it (B.L.); mariachiara.sbarra@asp-poli.it (M.C.S.); giovanni.ziarelli@asp-poli.it (G.Z.); sara.bagherifard@polimi.it (S.B.)

**Keywords:** bone-inspired, bone-mimicking, structural, aesthetic, garments, product design, architecture

## Abstract

The solutions provided through natural evolution of living creatures serve as an ingenious source of inspiration for many technological and applicative fields. Along these lines, bone-inspired concepts lead to fascinating advances in product design, architecture and garments, thanks to the bone’s exceptional combination of strength, toughness and lightness. Structural applications are inspired by the bone’s ability to resist fracture under a large spectrum of forces, while the high surface area and pore connectivity of bone architecture present exciting opportunities from an aesthetic point of view. Behind these inspirations, a disruptive common belief emerges: “down to the bone”, a journey in search of equality, universality and substantiality. Herein, we explore the current state of the art in bone-inspired applications in these fields, considering the two major categories of structural and aesthetic inspirations and discussing further technological developments.

## 1. Introduction

The ever-stringent design requirements on functionality, weight and durability, combined with the rigorous environmental, economic and social provisions, have urged the development of novel products that can offer optimized combinations of stiffness, strength, toughness and lightness all at reduced cost and environmental impacts. This increasing need for innovation is fortified by the development of biomimetic science, which deals with the study of biological processes in order to highlight innovative solutions to everyday problems [1]. Across years of evolution and natural selection, living creatures have resourcefully accomplished a wide range of captivating functions towards optimized performances as a requisite to survive, adapt and interact with their ever-changing complex natural environments [2]. This field specifically aims at understanding nature’s solutions as a mine of novel design ideas in order to exploit them to find concrete applications [3]. A source of inspiration is often the human body: this has recently led to the design of exoskeletons [4,5], as a substitute for amputated limbs [6] or as a support for gait rehabilitation, introducing the concept of biologically inspired mechanized creatures [7]. The peculiar characteristics of human figure are replicated not only as outward appearance, but specifically in terms of functional aspects [8].

In the recent years, there has been a significant advancement in this biomimetic field, especially thanks to the progress and versatility of additive manufacturing (AM) techniques, also known as 3D printing. AM in general consists of a vast group of promising technologies that are based on the deposition of successive thin layers of material upon each other for fabricating complex geometries; it offers the unprecedented possibility of realizing a wide range of personalized structures, starting from models obtained from computer-aided design (CAD) programs [9]. This progress has paved the way for exploiting intricate biomimetic designs to offer customized solutions in the fields of product design, architecture and garments with remarkable benefits both in terms of structural strength and aesthetics [10].

Among the objects of interest in bioinspired design, bones are a prominent candidate for many applications mainly due to their exceptional combination of stiffness, strength, toughness and lightness [11]. The multitude of benefits and functions induced by bone’s complex multiscale architecture, deeply detailed in Section 1.1, explain the growing diffusion of bone-inspired structures in a wide range of applications including product design, architecture and garments fields. Figure 1 testifies the increasing interest in bone-inspired concepts from 1950 to 2020 in a broad spectrum of fields.

It is to be noted that most of the available studies are focused on design and fabrication of innovative bone-inspired materials and particularly biomedical devices. Some impactful examples of bone-inspired materials [12] could find applications in biomedicine, i.e., membranes and coatings for scaffolds, as well as implants with a strong impact on disease management. Bone-inspired composites used in tissue engineering, indeed, show an enhanced fracture toughness and a remarkable strength–toughness balance [13] by implementing key bone microstructural toughening mechanisms [14,15,16,17].

However, more recently, fashion designers, architects and industrial designers have also started to appreciate and exploit the unique elements of bone structures in their respective sectors. Bone-inspired structures can be beneficial in these fields regarding both structural and aesthetic aspects. Structural applications are inspired specifically by the bone’s ability to resist fracture under a large spectrum of forces experienced during natural body movement, falls or collisions [18,19]. On the other hand, considering the high surface area, porosity and pore connectivity of bone architecture, the use of bone-inspired structures can present also interesting opportunities regarding aesthetic aspects, e.g., in terms of alternation of void’s and solid’s contribution to outline characterizing shapes. This is particularly of interest in fashion design where “bone-inspired” structures are mainly used for aesthetic purposes. Behind the increased interest in exploiting bone’s hierarchical configuration, there is a common belief, which the authors of this paper would like to summarize in the single concept of “down to the bone”. Besides the structural and aesthetic applications, bone is not a random choice of inspiration. Bone, which is able to sustain the weight and to provide structure and support to the whole body, is, above all, a strong symbol of essentiality, universality and equality. Therefore, the “down to the bone” concept explored by the designers and presented in the current paper, reminds us to go back to the core of objects, looking right at the nude skeleton as a pure substance. “Down to the bone” comprises the two aspects that are deepened in the present work, i.e., the structural applications that exploit bone’s multiscale exceptional mechanical characteristics and the aesthetic applications that capture bone’s peculiar hierarchical arrangement.

In this paper, we aim to explore the current state of the art in the application of bone-inspired concepts in product design, architecture and garments fields. To exploit the wealth of applications and opportunities that bone architecture can offer to these fields, it is necessary to detail the main concepts related to bone morphology and mechanical properties. Thus, we first provide a brief overview of bone architecture at micro-, meso- and macroscale. Then, we highlight the benefits and potential criticalities linked to the introduction of novel applications based on the concepts that mimic bone’s structure, providing a list of examples for each field regarding structural and aesthetic features. A critical eye is devoted to the comprehension of the conceptual aspects that lie behind the choice of bone-inspired structures, highlighting the relevance in conveying a message behind an individual product’s realization. Future advances and progress perspectives in these fields are also discussed.

### 1.1. Bone’s Morphology and Biomechanics as a Source of Inspiration

Bone is a hierarchically structured material, exhibiting distinct features with individual intrinsic mechanical properties at different length scales (Figure 2). There is a growing awareness that this hierarchical organization plays a significant role in determining bone’s bulk strength [20,21]. This is particularly relevant to this review’s purpose in order to comprehend how bone’s specific mechanical properties and intrinsic multiscale architecture act as a source of inspiration in different design fields.

At the macroscale, bones are organs belonging to the skeletal system. They perform relevant functions such as protecting internal organs, acting as a scaffold for muscles and for calcium storage, allowing movement and supporting body weight. At this gross anatomy scale, it is possible to distinguish structural features by naked eye and classify bones according to their shapes: long bones (e.g., femur, humerus), short bones (e.g., vertebra, carpal) and flat bones (e.g., frontal, rib, ilium). At the mesoscale, bones are categorized generally as cortical (compact) and trabecular (spongy) structures. The most evident difference between the two is associated with their different surface–volume ratios [22]. Cortical bone is found mostly in the diaphysis of long bones and surfaces of flat bones [23]. The porosity of cortical bone accounts for the presence of blood vessels, canals and cells, and it counts for about 5% of the cortical volume [24]. Compact bone is organized in osteons, parallel cylindrical structures made up of concentric layers (lamellae) packed together around a central canal (Haversian canal). Haversian canals are microscopic tubes with a diameter of around 200 μm [25] containing blood vessels and nerves. Neighboring Haversian canals are transversally interconnected by vascular channels named Volkmann’s canals. Trabecular bone, on the other hand, is a porous tissue with a surface–volume ratio much higher than the one in cortical bone [26]. This spongy tissue is predominantly found at the extremities of long bones and inner parts of the flat bones.

Trabecular bone consists of a network of interconnected rodlike and platelike structures named trabeculae, the architecture of which is highly variable and differs between species, anatomical sites and individuals. The structural features of the trabecular bone, including trabecular orientation, are not accidental but organized following the stress trajectories defined by the mechanical loads typically acting on bone, described by Wolff’s law [27,28]. When the environmental loads are changed by a trauma or a pathology, functional remodeling reorients the trabeculae to the new principal stress trajectories [27]. Bone’s structure is, in fact, designed to withstand compressive axial loads, the standard type of loading applied to the main bones of the body. At the microscale, it is possible to visualize lacunae, biconvex lens-shaped cavities where osteocytes reside, whose role is still under debate, having an effect on both strength and toughness. The lacunae are linked between themselves by dendritic channels called canaliculi, forming the lacuno-canalicular network where nutrients and waste can travel from one osteocyte to another [29].

On the macro-, meso- and microscale, bone shows intrinsic mechanical properties that make its hierarchical configuration particularly relevant from the viewpoint of the applications of interest in this paper.

## 2. Bone-Inspired Product Design

Bone inspiration has recently been widely applied in the product design discipline covering multitudes of applications. Here, two main categories of structural and aesthetic case studies are highlighted. A growing part of design research in this area is oriented towards replicating nature’s principles of conservation, such as optimization of material contributing to the so-called generative design [30]. This concept was born around 1970 in the automotive industry, but today it is a widely used tool among designers, especially thanks to the spread of commercial software such as Autodesk Inventor and Fusion 360. Using artificial intelligence (AI) algorithms and machine learning techniques, designers have the possibility of referring to solutions that are optimized for reducing development costs and time, material consumption and product weight.

### 2.1. Structural Applications

Among the benefits of bone-inspired structures, designers often take inspiration from the lightness of trabecular structures. Regarding the user experience, the exploitation of the trabecular structures’ lightness adds considerable value to the product itself, making it easier to handle and additionally improving the user’s comfort. Herein, we present a collection of furniture (Figure 3) in which trabecular lightness has been successfully traced and exploited exploring the concept of generative design. The typical load distribution applied during the testing phase of chairs (Figure 3a) is assessed and fulfilled in the suggested furniture.

Joris Laarman perfectly matches the “down to the bone” concept, developing an entire collection of bone-inspired furniture that includes a bone chair, a chaise, a rocker and an armchair with the underlying concepts of visual lightness and essentiality of the product. He started working on the Bone Chair [31] in 1998, taking inspiration from the algorithm implemented by Professor Lothar Harzheim [32], together with the International Development Centre of Adam Opel GmbH. This algorithm was first implemented to exploit topology optimization techniques for components of motor engines. In this collection, the algorithm was directly applied in order to combine multiple characteristics of bone’s structures (i.e., extreme lightness, load-bearing capability, porous arrangement), that are implemented to make the product comfortable, light and sufficiently stable. The Bone Chair is part of the permanent collection at Rijksmuseum (Amsterdam, The Netherlands), Museum of Modern Art (New York, NY, USA), Vitra Design Museum (Weil am Rhein, Germany), Centraal Museum (Utrecht, The Netherlands), Museum fur Kunst und Gewerbe (Hamburg, Germany). The Bone Chaise [33], another masterpiece of the collection, was fabricated by 3D printing. In this case, the author wanted to give a special icelike transparency to the chaise. This is the only piece of the collection for which a hand-made timber mold was used to test the design form before 3D printing. The Bone Rocker [34], part of the permanent collection of the Montreal Museum of Fine Arts, Québec, CAN, and the Museum of Fine Arts, Houston, TX, USA, was designed in 2007 and made of Noir Belge marble. It configures itself as the dark counterpart of the Bone Chaise, having a similar design and a high tendency towards visual lightness. It recalls the same concepts of topological optimization and loadbearing capability deeply explored in the Bone Chair. The third example of this collection that is surely worth mentioning is an armchair inspired by various bones [35]. Contrary to the previous pieces of the collection, the armchair is just molded and not 3D printed, even though the mold was obtained through the optimization algorithm already used for the chair and the chaise. The designer used a mixture of white Carrara marble powder and a casting resin to obtain this white porcelain-like armchair, which is part of the permanent collection of the High Museum of Art in Atlanta, GA, USA.

A similar approach was proposed more recently by Henrik Balzer [36] to create a “bone hammer”, implementing the principles of generative design (Figure 4a), which allows inputting design goals into the generative design software, along with parameters such as performance or spatial requirements, materials, manufacturing methods and cost constraints [37]. The software explores all the possible permutations of a solution, quickly generating design alternatives. In this specific case, the design focuses on trabecular structures, considering a combination of lightness and resistance as the key elements. Analogous principles are the basis of two other interesting case studies: the Cortex Cast (Figure 4b) made by the designer Jake Evill [38] and the Voronoi bicycle helmet (Figure 4c) designed by Yuefeng Zhou, Zhecheng Xu and Haiwei Wang [39].

Both Cortex Cast and Osteoid—the subsequent implementation proposed by Deniz Karasahin by adding a bone stimulation system [40]—represent two novel products that offer a user-centered perspective. They demonstrate how weight reduction can emerge as a rewarding value in the interaction with the end-user, especially in the case of prolonged and continuous use of the product where it often becomes synonymous of maneuverability and less user effort. The solution proposed by Evill [38], in particular, transforms a support of about 2 kg weight (depending on the type of fracture) to an exoskeleton’s weight of less than 500 g. The products inspired by trabecular structures exhibit both aesthetic and functional connotations. These concepts transform the aesthetics of the traditional plasters and fiberglass casts by suggesting a more fashionable design. Additionally, the knitted pattern provides breathable protection for the brace by limiting the risk of itching or other skin irritations such as skin burn, pressure sores or muscle atrophy, while allowing at the same time a customized distribution of the applied loads in the fracture zone. The localized increase in resistance is promoted using a flexible production process, which combines software and modern 3D scanning technologies. The process allows obtaining a product perfectly calibrated to the necessities of the patient, in terms of arm and fracture shape. Last but not least, this solution also permits the visual monitoring of the damaged area during the healing process while eliminating the typical complications related to the traditional casts.

The last case study analyzed in the category of product design for structural applications is the Voronoi bicycle helmet realized by Yuefeng Zhou, Zhecheng Xu and Haiwei Wang [39]; this design was granted “A design award” for safety clothing and personal protective equipment design category, representing a key concept based on multiscale bone inspiration. The aim of the designers of this alternative helmet was to suggest a safety product that would provide a valid combination of lightness and safety. The outer shell of the helmet is constructed from carbon Kevlar and fiberglass resin with a polystyrene lining, trying to mimic the architecture of Voronoi structures. The concept of Voronoi tessellation (often referred to as Voronoi diagrams) is a widely adopted procedure in mathematics for defining partitions starting from a finite set of nodes [41]. Assuming the existence of a set of fixed nodes in the plane, the Voronoi region associated with a point is the set with the smallest distance from the considered point among all other neighboring ones. Voronoi fractal structures are recurrent in biological tissues, especially in bone mesoarchitecture; they are used to generate representative parametric digital models that capture the microstructural features of real trabecular bones, and they are also implemented for creating 3D cellular structures with the faces and edges of the Voronoi cells considered as a pool of potential trabecular plates and rods, respectively [42,43]. There are also examples of the application of the Voronoi tessellation concept to bone microarchitecture [44]: in these studies, the partition is performed by considering a number of regions equal to the number of lacunae, with the aim to extract new information on the 3D morphometry of the lacunar network. The same procedure is applied in the Voronoi bicycle helmet: according to the Voronoi tessellation of the space, the sampling points of the internal bonelike structure are reorganized to define the integrated structure system of the helmet. The porosity of the outer shell is chosen in an adaptive and customizable way to show better stability against impact forces; this porous outer shell also contributes to maintaining the balance due to its extreme lightness.

### 2.2. Aesthetic Applications

Here, the main examples in the product design field where bone is mainly used as an aesthetic inspiration at different micro-, meso- and macroscales are presented (Figure 5). In particular, within this section three exemplifying case studies are reported: Vertebrae Staircase [45], Bone Chair [46] and Trabecular Lamp [47].

Although the products differ in terms of scale of interest and final application, in all of them the concept of bone inspiration is exclusively used for its aesthetic significance to shape the appearance of the object and to convey the “down to the bone” message through pure and naked design solutions. That is, in this category, the advantages typically associated with the adoption of a bone-inspired strategy regarding structural integrity are generally absent and not emphasized within the product function and its interaction with the user. The Vertebrae Staircase [45] (Figure 5a) is a clear demonstration of this concept. As extensively documented, the scale concept developed by architect Andrew McConnell in 2013 is entirely derived from the whale backbone structure; this is an inspiration that preserves an evident formal trace in the unit cell and in the assembled structure. Resistance and lightness are guaranteed in this case by the internal architecture of the modules, resorting once again to the support of generative design for weight minimizing. In particular, the combined use of internal metal plates and carbon fiber covering guarantees the required structural strength. On the other hand, the use of structural foam not only supports the carbon fiber cover but also ensures a good ratio between the lightness and the volume of the product. The concept of “Vertebra” is also used as a modular element in the fabrication of this object. However, this design is attributed mainly to the typical needs of industrial products, i.e., the simplification of production design and the assembly process, rather than to the desire for aesthetic continuity with the subject of inspiration.

The second example presented in this category is a bone chair designed by Daishi Luo and Zhipeng Tan from Mán-Mán Studio in 2019 [46]. This studio recently focused on bioinspired design products, with the purpose of reproducing harmony in shapes that can commonly be observed in biological phenomena. After designing lotus-inspired tables and chairs, Mán-Mán Studio developed a collection of furniture entirely inspired by the bone structure, such as a golden chair and a stool inspired by human basin and spinal column (Figure 5b). The designers get rid of extra furnishings, presenting a product that resembles a naked skeleton, free from any kind of adornment. The whole collection is made of copper and brass, which can be easily manipulated to obtain the desired shape. “Copper is alive, its plasticity is very high, and it is not what we always see”, the designers note on the choice of the material. Mán-Mán Studio has tried to combine the Gothic flavor of bone with smart use of plastic materials to come up with extremely fashionable design pieces. The 33-step chair first appeared at the 15th edition of Design Miami in 2019 in Basel, an international forum attended by designers, critics and collectors from around the world. The 85 × 45 × 50 cm^3^ chair is made of polished bronze.

Finally, the last example to be included in this section is the Trabecular Lamp (Figure 5c), developed by the designer Dimitri Vanless, in which the trabecular structure of the bone shapes the alternation of voids and solids of the product by creating a suggestive play of lights [47].

## 3. Bone-Inspired Architecture Design

When considering the use of bone-inspired design in architecture, the focus is generally on developing innovative structures that use new construction methods to improve material usage. In this field, bone features are also exploited as a source of inspiration from the viewpoint of both aesthetic and stability aspects.

### 3.1. Structural Applications

Among the natural bones available in nature, bird bones have particularly attracted the attention of architects thanks to their significant lightness and mechanical strength. Bird bones are extremely lightweight biological materials, and their structure can span multiple length scales while maintaining high stability [48,49,50]. They have an unusual combination of structural features, ranging from small- to large-scale multilayered organization, balancing their lightness with bending stiffness, torsional stiffness and strength of the whole structure [49]. Distinct bones of bird skeletons exhibit different densities, providing a more reasonable distribution of forces and better support for the specific activities of the corresponding species. This feature has been used in many design concepts, including the design of garments and buildings; in the latter category, bird bones have especially inspired roof architecture due to the large span and the strict requirements for lightness and structural stability, as schematized in Figure 6. Several of the following architectural applications are still at a preliminary concept level or are under assessment for realization.

Birdsong pavilion is a recent concept proposed by Exploration Studio [51], the design idea of which is “to place the material exactly where it is needed”, stressing the ideas of essentiality and substantiality that are predominant in the “down to the bone” concept. This design was based on detailed research, including modeling and 3D printing inspired by the crow’s skull, while its size is about 100 times the actual crow skull size. The bird skull consists of a very thin layer of bone material connecting the pillars, and at the same time, it has the advantages of saving material, offering an efficient structure for the dome and space framework. The special shape of this pavilion offers many cave-style spaces at different scales, which can be specifically used for musical performances and sound projection.

In standard designs that use traditional materials such as concrete and steel, the roof has always been considered to be heavy. However, to address this issue, the “Bone Inspired Pavilion” modelized by Andre Harris [52] proposed a concept, also in this case, based on the lightweight and strong bone structure found in birds’ skulls. The main purpose of this work is to generate a response structure that could resist external pressure to optimize material resources. Interestingly most bone tissues (especially in the skulls of larger songbirds) are composed of nondirectional spongy cells, which include vaporized cells that allow air to circulate between void areas, thereby reducing the total weight of the structure without affecting its strength.

Another interesting application inspired by bird skulls is the project of the Biomimicry Museum, that still hasn’t been built yet [53]. Under the condition of ensuring structural stability, the architects of this museum use, wherever possible, lightweight materials and reasonable openings so that the whole architecture can convey a light and bright feeling even when confined by huge roof and walls.

The last fascinating example where the structural and aesthetic functions of bone have been combined to create new architectural designs is the concept proposed for the Xinhee Design Center (Xiamen, China) [54]. In this context, while the building interiors represent the bone structure, with an atrium and six central frames spanning gardens and offices, the exterior is made of a polytetrafluoroethylene structure. This sun-shading envelope hangs over the vertical garden, giving the impression of a floating structure. Natural light and ventilation are facilitated by the large windows, which effectively mimic six petals. The bone structure acts as the frame for the whole building, providing wide spaces for the gardens.

### 3.2. Aesthetic Applications

An outstanding example of bone-inspired architecture is the new building of the Royal College of Physicians, the “Spine” [55]. Situated in Liverpool (Liverpool, United Kingdom), the 13-floor construction is set to be completed in 2021. The name “Spine” is associated with the distinctive staircase in the main entrance, which resembles the shape of vertebrae. Moreover, the building is surrounded by a Voronoi pattern of lights (Figure 7) that mimics the trabecular structure of the bone, with the aim of illuminating the whole building. In this context, the “down to the bone” concept emerges once again, as a reminder to go back to the essential units, represented by the trabecular pattern of lights. Additionally, the presence of the “Spine” staircase enhances the willingness to look exactly at the nude vertebral column.

The most well-known case studies of bone-inspired architecture in aesthetic aspects could probably be the works of the celebrated architect, Gaudi. In Casa Batlò in Barcelona (Spain), as a renovation project on existing architecture, Gaudi used bone-inspired elements for decoration of the facades and the roof (Figure 8). The message conveyed here by means of the bone-inspired design assumes both symbolic and social implications [56]: the purpose of the exterior design is to represent the battle between Saint George and the dragon, a tribute to the Catalonian culture, in which the skulls are piled up in a tower of defeated enemies. In this building, the facades are also engraved with trencadís (broken plates of ceramics), which contrast the pale tone of the bones and create distinctive cultural connotations [57]. Another case of Gaudi’s work is the “La Sagrada Familia”, again in Barcelona, in which the bone-inspired elements have both aesthetic and structural functions. Gaudí used the bone features for the interior load-bearing structure where some bone-inspired elements were left uncovered to serve as interior decorations [57].

Another interesting example is proposed by Joe MacDonald of Urban A&O Architecture LCC who suggests “bone wall” as a creative solution of load-bearing structure [58]. In this facade, all the 72 cells that made up the wall were based on one single generative unit cell, recalling once again the concept of microscale bone tessellation. Facades made with a traditional structure and traditional materials have very limited options of openings and decorations due to the structural requisites associated with load distribution. Thus, this installation has an aesthetic feature, but it could further be used as a structural element in the design of building facades.

## 4. Bone-Inspired Garment Design

Bone structure in all its complexity is a unique source of inspiration for fashion designers, especially for the development of innovative structures that exploit bone’s multiscale porosity.

### 4.1. Structural Applications

Bone’s peculiar mechanical properties and intrinsic symbolic implications have been considered in the garment field for structural applications, typically for items that are subjected to compression such as footwear. There are two notable representative examples of bone-inspired structures used in the design of shoes. The first example is the “Melonia Shoe” [59] (Figure 9a), a creation of the fashion designer Naim Josefi and the industrial designer Souzan Youssouf, displayed at Mercedes-Benz Fashion Week (Stockholm 2010). The second case is the “Biomimicry Shoe” [60] that is a product of collaboration between fashion designer Marieka Ratsma and architect Kostika Spaho (Figure 9b).

Customization is a main feature of the “Melonia Shoe”, which is claimed to be the first-ever 3D printed couture shoe. The vision of the designers for these shoes with customizable design includes scanning the user’s feet and generating a 3D model of the shoe through Rhinoceros software [61], a commercial 3D computer graphics and CAD application software developed by Robert McNeel & Associates. The designer exploits the advantages of CAD modeling [62,63] to exactly fit the shape of the specific feet and then 3D prints the shoe through a layer-by-layer production strategy.

The concept of 3D printed shoes has attracted the attention of multiple well-known brands, such as Adidas, Reebok and New Balance, that are incorporating AM techniques into their fabrication line and trying to industrialize this approach to reach mass customization. The main benefits obtained through the application of AM in shoe fabrication are in terms of ergonomics, lightness, customization and optimal foot control during usage, which is particularly important for sports shoes.

As represented in Figure 9, the design of both these high-heel shoes is inspired by the trabecular bone morphology, even if they are characterized by different design elements. The design of the “Biomimicry Shoe”, in fact, exploits mainly the lightness and the specific morphology of a highly differentiated bone, resulting in efficiency, strength, resistance to compression and elegance, typical of the “down to the bone” concept. In this project, the back part of the shoe is specifically inspired by the shape of a bird’s cranium, whereas the front part recalls trabecular bone architecture.

The use of a network of trabeculae and rods as the reference pattern leads to a less dense, lighter and more flexible design that represents, at the same time, an adaptable architecture, also in “Melonia Shoe”. On the other hand, the designers of this shoe set its main objective as promoting sustainability; in this case, the bone architecture is used to minimize the quantity of waste material, ensuring at the same time perfect structural stability. The designers abandon the traditional linear economic model, based on the exploitation of finite environmental resources, in favor of the circular economy. This leads to the study of materials that could be re-evaluated in the production and consumption cycle. An example is nylon, a synthetic fiber composed of polyamides known for its strength and durability with a degradation time that, until recently, was around 40 years [64]. New recycling technologies have been developed with the aim of exploiting the high mechanical properties of nylon while reducing its environmental impact. The choice of recyclable nylon as the material of use reflects the designers’ sustainable perspective. Table 1 provides a comparison between the mechanical properties of nylon and the materials typically used for shoe fabrication. It is interesting to point out that the trabecular bone inspiration for the “Melonia Shoe” perfectly fits the choice of nylon. The combination of specific material properties (increased modulus of elasticity and reduced density) and spatial arrangement leads to an about 17–62% lighter and 100 times more resistant structure [65].

### 4.2. Aesthetic Applications

In addition to structural applications, purely aesthetic design concepts have also considered bone structure as a valuable source of inspiration. In the clothing field, the introduction of this concept is strictly linked to the possibility of realizing 3D printed garments, creating a new beauty concept through the combination of disruptive sartorial technologies and bone-inspired design (Figure 10).

A stylist celebrated for her capability to use the latest technologies to mimic human tissues is Iris van Herpen, who, in collaboration with architect Isaïe Bloch, has designed the “Skeleton” dress (Figure 10a) using 3D printing technologies [66]. This item belongs to her Capriole Collection presented in Paris Haute Couture Week in 2011. This white polyamide dress tries to mimic the structure of the bone at both the macro- and mesoscale. As stated by the designer, this dress represents the pure and naked body of all humans in the moment of the free fall when each element of the skeleton expands in a different direction.

Another fashion designer who tried to blend 3D printing and bone structure is Tang Xiao, who at 2016 Wuhan Textile University Fashion Week showcased three dresses made of bonelike structures (Figure 10b) that were also inspired by the Sirens of Greek mythology [67]. Despite the novel aspects implemented in this design, the stylist highlighted some criticalities. The first issue is the high cost of around USD 8000 in the creation of these dresses. The other issue deals with the conformability of the dresses, since the pieces realized through 3D printing were made of rigid materials, contrary to the commonly used materials for clothes, which fit the continuous changes of body shape; therefore, it is expected that this dress line would most probably remain limited solely to aesthetic exhibitions.

There are also examples of jewelry inspired by the natural shape of the human bone, where the focus of the designers has also been to exploit the multiscale structure of bone. “Vertebrae” (Figure 10c), the necklace designed by Molly Epstein, is inspired by the macroscale shape of the human spine [68]. The message that the designer intended to communicate metaphorically through the anatomical appearance of this adornment is that humans all have the same blood running through their veins and bones under their skin. The message is to urge the observer to go beyond the ornament itself, enhancing once again the “down to the bone” metaphorical principle.

Other examples of jewelry design using microscale bone structure are some pieces of the “Cell Cycle” collection created by Nervous System Company (Figure 10d), including 3D printed rings, pendants and bracelets available in different materials such as sterling silver, stainless steel, gold-plated steel, polyamide or silicone [69]. The peculiarity of these items is that the customers can collaborate with designers over a web app tool to generate customized jewelry pieces. The variety of possible designs and the complexity of the geometries obtainable with this tool make this bone-inspired concept particularly prominent and customizable.

## 5. Discussion

The three different scales of the bone structure appear to be mutually employed within the investigated fields of product design, architecture and garments in both aesthetical and structural applications (Figure 11). In particular, it is interesting to note that the macroscale architecture of bone has a wide range of applications, although they are exclusively of an aesthetic nature. Over time, the macroscale has in fact assumed strong cultural and symbolic connotations that have made it an expressive and strongly evocative element for designers and artists. A more careful analysis shows a clear prevalence of interest in mesoscale features: among the analyzed case studies, 11 of them refer to the mesoscale bone characteristics, 6 correspond to macroscale bone characteristics and 3 relate to the microscale aspects of bone structure. The use of the mesoscale is indeed an optimal compromise between the advantages derived from the application of a bone-inspired structure and the density of the lattice. It is no coincidence that this solution finds greater applications in the field of architecture where the high dimensions of the architectural works allow the reticle to be reproduced on a large scale and make it easily reproducible even with the use of traditional construction techniques.

Within the microscale-inspired structures, the only case study representing a realized product is the Voronoi bicycle helmet. However, at present, it is not commercialized, and the bicycle helmet still represents an exclusive concept. This observation underlines the production difficulties at the microscale dimension when applied outside the architectural field, where the features are rescaled.

Among the case studies inspired by bone’s microscale features, the Voronoi tessellation is a recurrent concept. Apart from the abstract mathematical notion of partitioning the plane, Voronoi decomposition has been recently studied by Mader et al. [70] in the context of bone microscale features, where space tessellation was used to identify neighboring elements, considering the distribution of bone microscale components (such as lacunae) in the 3D space. The same reasoning is applied to the Spine [71], one of the mentioned aesthetic applications of bone-inspired architectures, which has a facade decorated with a striking Voronoi pattern through the help of light games. Apart from the aesthetic function of the pattern, the external microscale bone-inspired outline assists the solar control of the building, implemented through ad hoc external glasses to offer optimum temperature comfort and adequate sun exposure. In the inner part of this building, the Voronoi structure is implemented in the ceilings, generating an alternation of empty and full spaces, exploitable for the positioning of pending lights. Another interesting case study where Voronoi design is involved is the bicycle helmet mentioned in Section 3.1 that took advantage of the Voronoi structure’s optimization in its three-dimensional shape to offer considerable lightness combined with load-bearing characteristics. The specific design of the helmet offered a homogeneous distribution of weight and resistance. Another fascinating feature derived from the high porosity induced by Voronoi partitioning is that the morphological disposition can be easily parametrized, 3D printed and customized.

The AM technique is another often-exploited topic in this review that is worth mentioning; AM is a rapidly expanding area owing to its capacity to fabricate parts of intricate geometries with customized features for a wide range of applications. AM clearly emerges as a dominant trend among bone-inspired products; this is particularly more tangible in the field of garment and product design, presumably due to the scale of reproduction of such products, far smaller than the ones of architectural works. In addition, the increasing availability of processable materials has encouraged the spread of AM among designers and companies by offering them significant flexibility and new opportunities in terms of process innovation and expression. In the “Cell Cycle” case study designed by Nervous System the user has the possibility to live a fully customized experience, perceiving the coevolution, namely the continuous change of the design concept in the light of her or his exploration of possible solutions [72]. The user becomes the creator of her or his own jewel by modifying the 3D model available on the Nervous website, without limiting the outputs of conceptual design [73]. Further forms of customization and direct interaction with the end-user can be found in two other case studies analyzed in the present review: the Melonia custom-made shoes and the 3D printed Cortex Cast. In such cases, however, customization manifests itself in the form of “tailor-made” products. Joining the forces of AM techniques and new 3D scanning tools has enabled the production of solutions literally built on the “shapes” and the needs of the user. The recent progress suggests that the process flexibility and high customization possibilities of AM technologies will be the key factors for their widespread in these fields in the near future [74,75,76]. A schematic representation of the 3D printing procedure for the mentioned cases is provided in Figure 12.

Beyond the common technical aspects in bone-inspired product design, architecture and garments, an interesting observation emerges from the psychological field. There is a marked tendency of the designers in all the three mentioned fields to convey a personal message through their bone-inspired creations, apart from all the aesthetic or structural purposes. Through an object or an ornament, the designer can communicate an ideology, but, in addition, by buying or wearing the object of interest, one can become a symbol or a spokesperson for that idea. For instance, the “Skeleton” dress created by Iris van Herpen is the triumph of the mentioned “down to the bone” ideology: a representation of a skeleton deprived of any ornament that can be interpreted as a symbol of equality; all humans, devoid and free of superfluous ornaments, are characterized by the same common core. The same concept of equality is hidden in the necklace “Vertebrae” by Molly Epstein. The cited message is a common thread among almost all the product and fashion design and architecture works, which will have an impact on the design transformation [77] and social turn [56] of the next years.

To sum up, this paper provides a general overview of various aspects of bone-inspired designs where bone’s resistance to compression, lightness and peculiar hierarchical architecture have been exploited as a source for the generation of extraordinary concepts and products in product design, architecture and garments. Fascinating case studies highlight the significant potential of bone structure and morphology to be exploited both for structural and aesthetical purposes in these sectors. The actual state of technology suggests that, in particular, mesoscale bone-inspired structures have been extensively applied in product design, architecture and garments by harmoniously combining structural and aesthetical necessities. With the advancement of AM technologies and improved prototyping resolutions, it is expected that bone morphological features can serve as sources of inspiration in all their multiscale aspects, including the microscale, which has been less exploited due to the current technological limitations.

## Figures and Tables

**Figure 1 materials-14-04226-f001:**
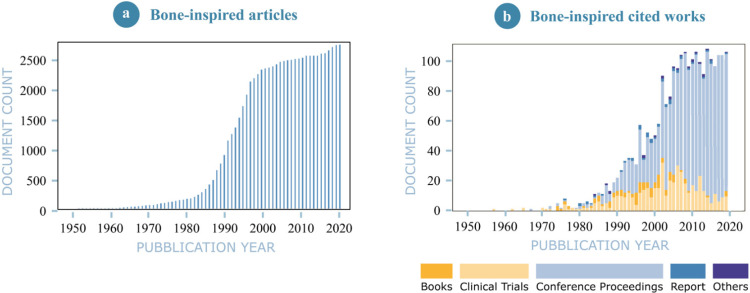
(**a**) Graphical representation of the trend in journal publications on bone-inspired design from 1950 to 2020. (**b**) Other bone-inspired studies categorized as books, clinical trials, conference proceedings, reports, patents, etc.

**Figure 2 materials-14-04226-f002:**
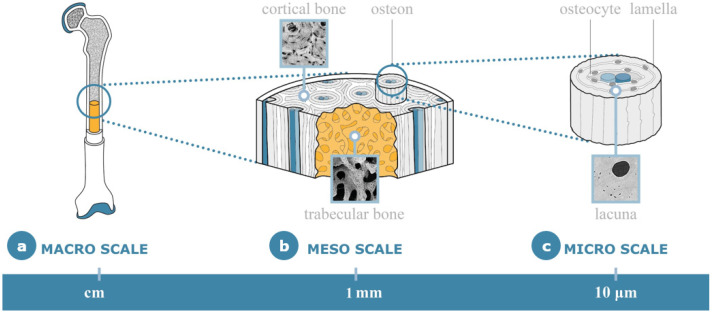
Schematic representation of bone multiscale hierarchical architecture. (**a**) Macroscale represents the whole bone features, while at the mesoscale (**b**), it is possible to distinguish cortical and trabecular structures. At the microscale (**c**), lacunae, small cavities where osteocytes reside, can be noted.

**Figure 3 materials-14-04226-f003:**
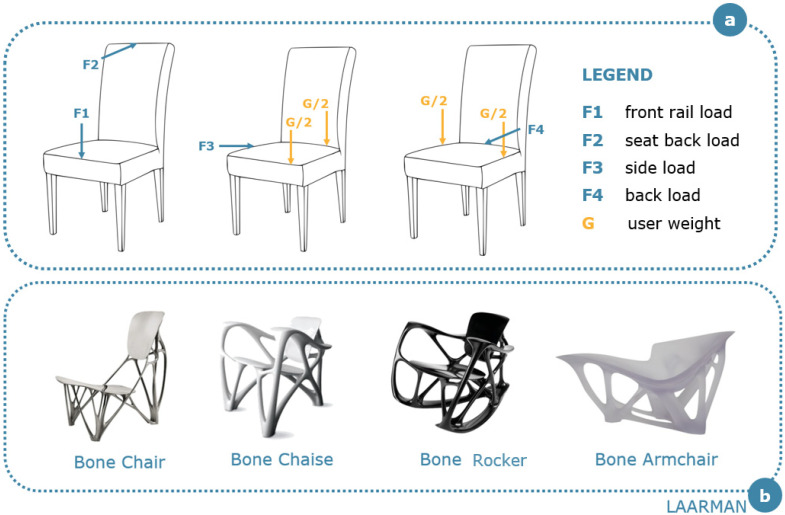
(**a**) Schematic representation of the loads applied during the testing phase of chairs. The user weight is considered as an additional variable. Several configurations are considered, such as the loading of the front rail (F1) and the seat back (F2), the side loading (F3) and the back loading (F4). (**b**) Laarman’s bone-inspired chair collection.

**Figure 4 materials-14-04226-f004:**
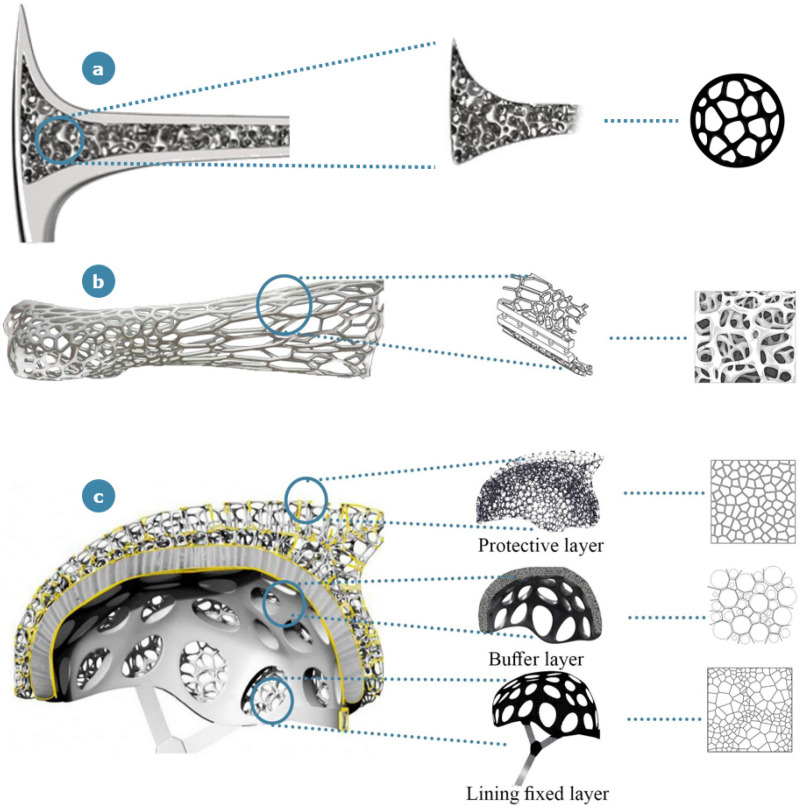
(**a**) “Bone Hammer” by H. Balzer; (**b**) “Cortex Cast” by D. Karasahin; and (**c**) “Voronoi bicycle helmet” by Y. Zhou, Z. Xu and H. Wang. In (**a**,**b**), the source of inspiration is represented by the mesoscale features of bone’s hierarchical structure, while in (**c**), the idea of three protective helmet layers is derived from bone’s microscale.

**Figure 5 materials-14-04226-f005:**
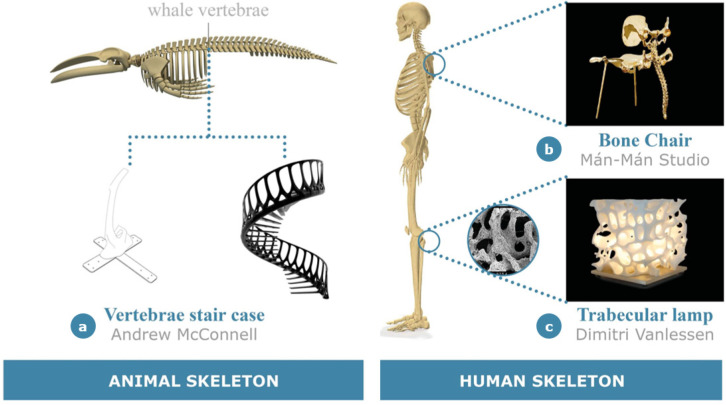
Aesthetic design applications: (**a**) Vertebrae Staircase by A. McConnell; (**b**) Bone Chair by Mán-Mán Studio; (**c**) Trabecular Lamp by D. Vanlessen.

**Figure 6 materials-14-04226-f006:**
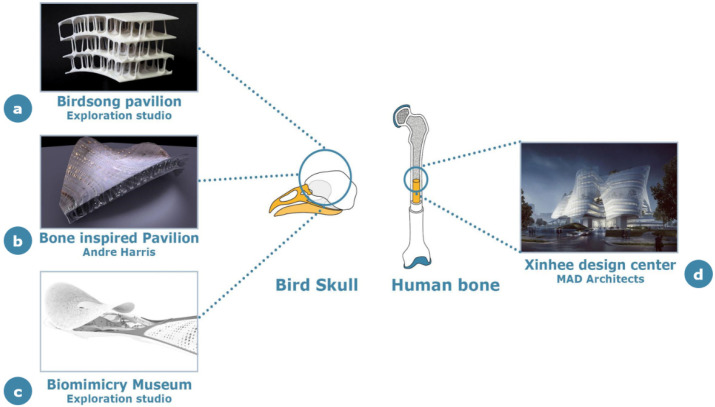
Structural applications of bone-inspired concepts in architecture: (**a**) Birdsong Pavilion by Exploration Studio, (**b**) Bone Inspired Pavilion by A. Harris, (**c**) Biomimicry Museum by Exploration Studio and (**d**) Xinhee Design Center by MAD Architects.

**Figure 7 materials-14-04226-f007:**
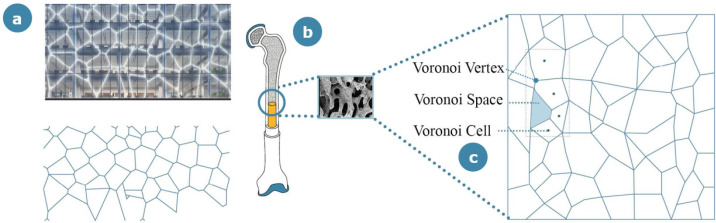
Bone-inspired aesthetic application in the architecture field: (**a**) the Spine building; (**b**) schematization of the concept: the facade of the building resembles bone micro-architecture, (**c**) generating a Voronoi pattern, characterized by vertexes, cells and spaces.

**Figure 8 materials-14-04226-f008:**
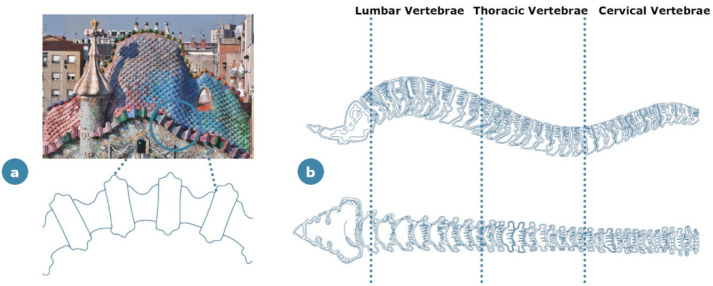
Aesthetic bone-inspired application in the architecture field: (**a**) Casa Batlò in Barcelona (Spain); (**b**) schematization of the concept: the cornices are arranged in a geometry that resembles human vertebrae.

**Figure 9 materials-14-04226-f009:**
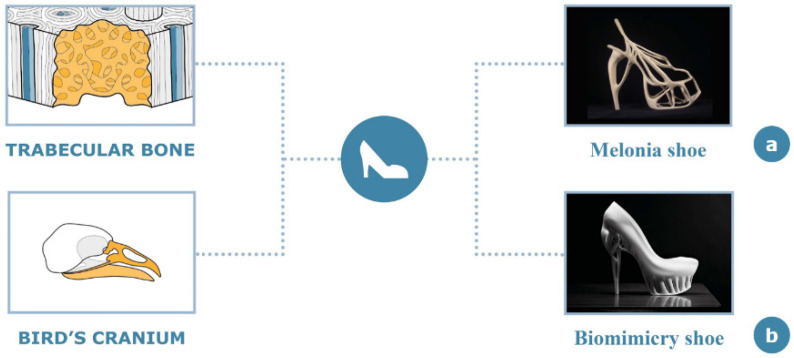
Genesis and inspiration of “Melonia Shoe” (**a**) and “Biomimicry Shoe” (**b**).

**Figure 10 materials-14-04226-f010:**
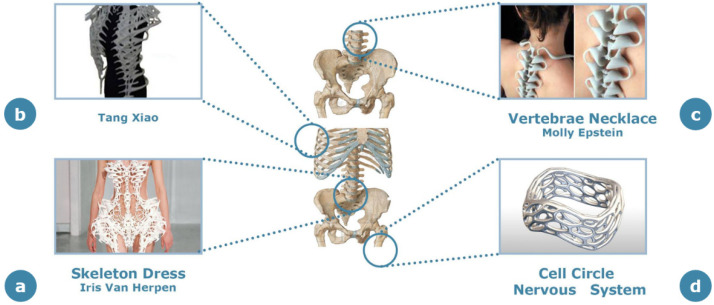
Schematic representation of different bone-inspired aesthetic garments in the garment design field. The site of the human skeleton used as a source of inspiration for each product is highlighted with blue circles. (**a**) “Skeleton” dress by Iris van Herpen; (**b**) bone-inspired dress by Tang Xiao; (**c**) “Vertebrae” necklace by Molly Epstein; (**d**) “Cell Circle” by Nervous System Company.

**Figure 11 materials-14-04226-f011:**
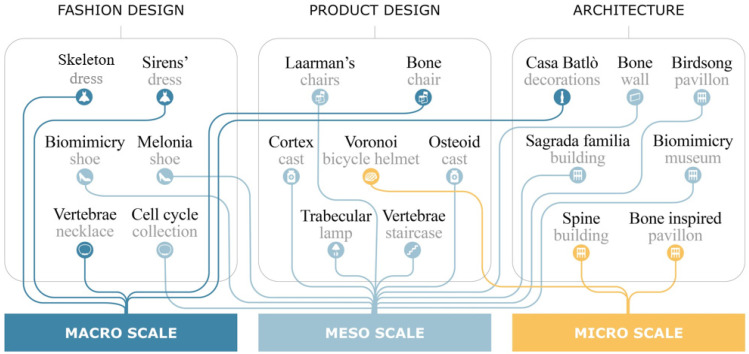
Existing interconnections between bone hierarchical scales and fashion, design and architecture application fields.

**Figure 12 materials-14-04226-f012:**
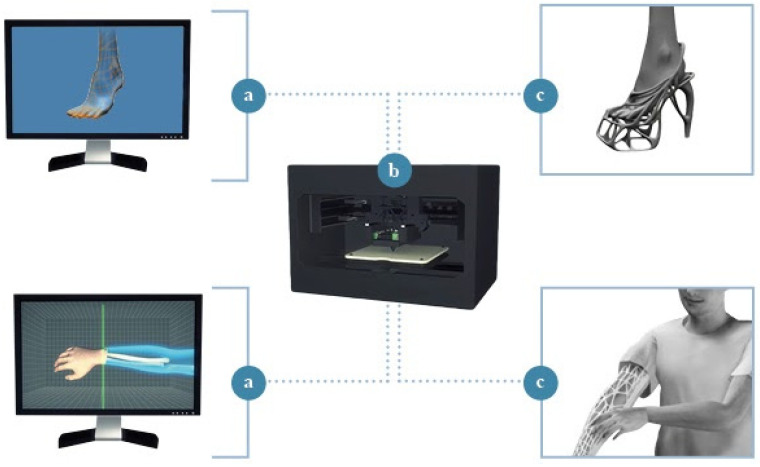
(**a**) The shape of the subject’s foot/arm is scanned (**b**) A 3D model of a shoe/cast is generated to exactly fit the shape of the foot/arm and is 3D printed (**c**) A customized final product is obtained.

**Table 1 materials-14-04226-t001:** Comparison between the mechanical properties of nylon used for “Melonia Shoe” and other materials commonly applied for footwear [65].

	Nylon	Leather	Rubber	Polyester
Young’s Modulus (MPa)	2600–3200	100–500	1–25	2100–4400
Yield Strength (MPa)	50–95	5–10	20–30	33–40
Tensile Strength (MPa)	90–165	20–26	22–32	41–90
Density (g/cm^3^)	1.14	2–3	1.52	1.38

## Data Availability

No new data were created or analyzed in this study. Data sharing is not applicable to this article.

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
