# Peer review of "Down to the Bone: A Novel Bio-Inspired Design Concept"

_materials, 2021, doi:10.3390/ma14154226_

Round 1

Reviewer 1 Report

This paper proposes a new bio-inspired mechanical design concept; the inspiration is very interesting and has been implemented appropriately. The findings are relevant to the state-of-the-art and could be beneficial to the community of bio-inspired design and engineering.

However, a more detailed literature review on mechanical bio-inspiration is recommended. In particular, it is important to provide readers with a more profound background on the previous studies on mechanical bioinspiration from living creatures for human/humanoid design applications. The following references are suggested, among other references that the authors may wish to cite to improve their literature review:

  • DOI: 10.1007/s00332-016-9305-x
  • DOI: 10.1126/science.aam9075
  • DOI: 10.1063/1.4964136

I recommend this manuscript for publication provided that the abovementioned revision is made to it.

Author Response

We would like to thank the reviewer for the detailed comments and for the positive feedback on our work.

The suggested interesting deepening concerning mechanical bio-inspiration for humanoid design applications has been implemented and highlighted in yellow in the manuscript. The topic represents in the authors’ opinion an added value to the discussed aspects for two main reasons that have been considered in the text:

  • Human body as a whole is often a source of inspiration. In this review we focused on bone for its exceptional mechanical properties and peculiar architecture, but fascinating applications try to emulate entire limbs, as it happens for exoskeletons.
  • Humanoid design combines the two main aspects that we discussed in the review: structural and aesthetic applications. The peculiar characteristics of human figure are indeed replicated not only as external appearance, but specifically in terms of functional aspects. The example of the biologically inspired lower limb is particularly interesting in terms of the structural parameters’ optimization, that directly derives from the analysis of the human posture.

Reviewer 2 Report

In the article entitled "Down to the bone: a novel bio-inspired design concept" authors presented bone-inspired design in architecture and clothes. The article is written nicely, easy to read and understand the presented idea. I point to two places where corrections should be provided:

line 144 I suggest changing "typical type" into something different like "standard type" or "frequently used type"
line 199 "...High museum of Art..." should be "...High Museum of Art..."

The overall impression is relatively poor in terms of Materials journal scope. It is a scarcity of scientific and purely physics-related information like numerical values of weight reduction of created products. Even information on page 13 with Table 1 describing materials properties refers just to general properties and not point to any gain related to bone-inspired design. I suggest that authors submit the article to a journal more related to art and design.

Author Response

We would like to thank the reviewer for the precise review process and all the useful provided comments.

The suggested corrections have been implemented in the manuscript and Table 1 has been contextualized to emphasize the advantages that the use of nylon provides to bone-inspired shoes. More in depth, it is highlighted that the high elastic modulus and the low density of nylon when compared to other materials commonly applied for footwear make nylon a suitable choice. Additionally, some comments have been added regarding the trabecular architecture of the shoe that enhances the concepts of lightness and resistance that are both in line with the material’s selection.

The review explores the novel bio-inspired design concept called “down to the bone”, addressing bone’s specific mechanical characteristics and morphological features and how they are specifically implemented in structural and aesthetic applications in different fields. A critical comparison is performed, analyzing how multi-scale bone characteristics have been readapted and applied to product design, architecture and garments. Additionally, for each mentioned structural application, a comparison with the traditional mechanical features of the corresponding product is reported. As an example, the disruptive innovation provided by Laarman’s chair in terms of lightness and resistance is contextualized with the traditional process of chair’s testing. Another case study is the one of the “Melonia Shoe”, where a critical evaluation between the traditional materials and the chosen recyclable nylon is provided. Another mechanical topic that emerges from this review is the transversal use of additive manufacturing technique, that specifically allows to obtain custom products. On the aesthetic side, this review considers how bone multi-scale hierarchical architecture is exploited, deepening how the mathematical notion of Voronoi tessellation is transversally applied. Thus, we believe that covering these innovative aspects inspired by the structural and aesthetic features of natural bone, could be in line with the aim and scope of the Material Journal.

Reviewer 3 Report

This study dealt with the design of various products, architecture, and clothing based on the anatomical structure of bones. Overall, the bone-inspired design was well illustrated with appropriate examples. In particular, the figure composed of neat graphics is impressive. However, it would also be nice to cover the latest bone-inspired materials developed to treat various diseases.

Author Response

We would like to thank the reviewer for the provided comments and for the positive feedback on our work. In Figure 1 we have reported that there is an increasing interest in bone-inspired concepts with several applications in a wide variety of fields. More in depth, we have added in yellow in the manuscript some interesting aspects related to bone-inspired composites that could be applied to the design of membranes, scaffolds and biomedical implants. It is relevant to point out that, in the biomedical field, bone is chosen as a biomimetic model for its amplification in toughness with respect to its constituents and for its remarkable strength-toughness balance, as underlined in the manuscript. The practical realization of these bone-inspired composites is possible due to the versatility of additive manufacturing technique, as underlined in the discussion of the review.

Reviewer 4 Report

This paper firstly introduces the Bone Structure from the micro, meso and macro perspectives, which shows the characteristics of high strength, toughness and lightweight of the Bone Structure. Based on the inspiration provided by Bone Structure, the author discusses the existing application cases in product design, architecture andgarments from two aspects of structure and aesthetics. At the same time, it puts forward a subversive concept "downto the bone", which is devoted to the study of its equality, universality and substantiality.

Over all, the manuscript is well organized, rich in content, and logical in accordance with the theme. There are still some shortcomings in this paper, and the following suggestions are put forward to help improve the manuscript:

  1. The structure of Introduction is a bit disordered while thinking jump, the hierarchical architecture is metioned in the third paragraph while it is explained at the end of Introduction, making readers understand unclearly;
  2. Line 169 What is the specific method of "generative design" mentioned in the first paragraph of the second part? Why is it proposed separately? How does it relate to the products mentioned below?
  3. Line 193 What is the relationship between the algorithm and the bone chair? Please explain it  appropriately;
  4. P5 Fig3 The Bone Rocket shown in Figure 3 is not introduced in this paper so that the image and text do not correspond;
  5. Line 479 In Table 1, why is "MeloniaShoe"perfectly fits the choice of nylon material?
  6. P16 The summary of AM technique in Discussion is too long, which can be briefly described or removed;
  7. For the concept of "down to the bone" in this paper, it is only a piece of writing at the beginning and end, which is too scribbled to let readers understand the meaning of the theme.

Author Response

We would like to thank the reviewer for the detailed comments and for the positive feedback on our work.

All the proposed improvements have been implemented in the manuscript as follows:

  1. In order to avoid any kind of confusion in the introduction, the mention of bone’s hierarchical structure has been removed from the third paragraph of the introduction and a specific cross-reference to sub-section 1.1 is inserted. Sub-section 1.1 is entirely devoted to a detailed explanation of bone’s complex multi-scale architecture, that is recognized as the pivotal reason behind the increasing interest in bone-inspired products.
  2. The concept of “generative design” is introduced in the first part of the section 2 as a recurring theme in all bone-inspired product design examples. It consists of exploring the power of computational techniques and artificial intelligence in order to identify the optimized solution with respect to the defined constraints. As regards the considered design applications, the generative design is exploited both in structural and aesthetic applications. An additional reference to this topic is added and highlighted in yellow in the text. The crucial reason behind the use of generative design is linked to the realization of lightweight structures, from both a mechanical and a visual point of view. Some examples of this concept are Laarman’s chairs, that present a clear tendency toward lightness and essentiality of the product. In addition to this, we have mentioned the “Bone Hammer” by Henrik Balzer, that considers a combination of lightness and resistance as the key parameters for the final solution. As regards aesthetic applications, it is worth citing the “Vertebrae” staircase by A. McConnell. It is reported in paragraph 2.2 and in Figure 5 as an example of bone-inspired design concept, that resorts to the support of generative design for the conceptualization of the idea.
  3. More explanation on the direct link existing between Laarman bone chair and Harzheim’s algorithm is added to the manuscript and highlighted in yellow. More in depth, the algorithm offers a topological optimization solution and it is deeply explored by Laarman when looking for an optimal design for the bone chair, while combining multiple characteristics offered by bone’s architecture, such as lightness and load bearing capability.
  4. An additional explanation of Laarman’s Bone rocker is added to paragraph 2.1 and highlighted in yellow. The Bone rocker recalls the same concepts of topological optimization and loadbearing capability deeply explored in the Bone chair.
  5. The choice of nylon for the realization of the “Melonia Shoe” has multiple reasons, as described by its designer: first of all, the combination of nylon’s specific material properties (increased modulus of elasticity and reduced density) allows to design a lighter but at the same time more resistant product, with respect to the traditional materials’ options (i.e., leather, rubber, etc.) for shoes’ fabrication. Additionally, we have pointed out another crucial aspect that endorses the choice of nylon: new recycling technologies have exploited the high mechanical properties of nylon, while reducing its environmental impact. The choice of recyclable nylon as the material of use reflects the designers’ sustainable perspective.
  6. The AM technique’s advantages have been summarized in the discussion section, keeping just the central aspects of process flexibility and high customization possibilities, as suggested by the reviewer.
  7. The novel and disruptive metaphor of “down to the bone” is better clarified in the whole paper and highlighted in yellow through the manuscript. “Down to the bone” comprises the two aspects that are deepened in the present work, i.e., the structural applications that exploit bone’s multi-scale exceptional mechanical characteristics and the aesthetic applications, that capture bone’s peculiar hierarchical arrangement.The concept explored by the designers and presented in the current paper, reminds to go back to the core of objects, looking right at the nude skeleton as a pure substance. Several concrete applications of the metaphor are presented in the manuscript: Laarman’s chairs collection, Vertebrae staircase by A. McConnell, the Spine building, different bone-inspired shoes, “Vertebrae” necklace by M. Epstein, etc.

Round 2

Reviewer 2 Report

Dear authors, 

Thank you for all provided explanations, but despite them, I'm afraid I have to disagree that you present extensive mechanical and physical aspects of the bone mimics design. Your additional information in Table 1 is not relevant as presented values are general for materials, not for design. You may produce these shoes from any other polymer. Where this 100 times more resistant value come from? What was the test, or at least provide a source of this information? I do not assess aesthetic aspects of bone mimics design as I am not an artist, and that will not be possible for me from a purely scientific point of view. I still support my previous decision that the article is more suitable for more art-related journals.

Author Response

We would like to thank the reviewer for the comments and suggestions.

As regards Table 1, it aims at comparing the mechanical properties of traditional materials that are commonly used for shoes’ fabrication and the “Melonia Shoe” designer’s choice for recyclable nylon. Nylon’s specific material properties (increased modulus of elasticity and reduced density, as reported in Table 1 allows to design a lighter but at the same time more resistant product, with respect to the traditional materials’ options (i.e. leather, rubber, etc.) for shoes’ fabrication. “Melonia Shoe” couldn’t be produced from any other polymer, because the choice of recyclable nylon reflects the designers’ sustainable perspective, aiming at directly implementing the concepts of circular economy in their product.

Regarding the specific values for nylon reported in Table 1, the source for all the information is the “Materials data book” from the Cambridge University, Engineering Department, that provides an broad database of several mechanical characteristics of different materials. The 100 times higher resistance of nylon is derived from a qualitative comparison between the yield strengths values of nylon and the traditional leather. The respective reference is cited in the manuscript.

The aim of this section, devoted to structural applications of bone-inspired garment design, is to introduce several products that are characterized by an evident inspiration to bone architecture, exploiting their exceptional mechanical properties. For this reason, bone’s high resistance under compression is taken as a model for the design of footwear, typically subjected to a compressive loading scenario during the stance and walking phase. No specific test is presented for each one of the bone-inspired shoes, since these products are still not commercialized.

As a general insight, the present review offers a wide panorama of both structural and aesthetic applications of bone hierarchical structure, with a special focus on application in fields that are less studied. The aim is to underline the significant potential of geometrical and mechanical aspects of bone structure to inspire a vast range of applications.

Reviewer 4 Report

The manuscript was revised well. No any special comments was proposed.